# Complete Utilization of the Major Carbon Sources Present in Sugar Beet Pulp Hydrolysates by the Oleaginous Red Yeasts *Rhodotorula toruloides* and *R. mucilaginosa*

**DOI:** 10.3390/jof7030215

**Published:** 2021-03-17

**Authors:** Luís C. Martins, Margarida Palma, Angel Angelov, Elke Nevoigt, Wolfgang Liebl, Isabel Sá-Correia

**Affiliations:** 1iBB—Institute for Bioengineering and Biosciences/i4HB—Associate Laboratory Institute for Health and Bioeconomy, Instituto Superior Técnico, Universidade de Lisboa, 1049-001 Lisbon, Portugal; luismcmartins@tecnico.ulisboa.pt (L.C.M.); margarida.palma@tecnico.ulisboa.pt (M.P.); 2Department of Bioengineering, Instituto Superior Técnico, Universidade de Lisboa, 1049-001 Lisbon, Portugal; 3TUM School of Life Sciences, Technical University of Munich, 85354 Freising, Germany; angelov@tum.de (A.A.); wliebl@wzw.tum.de (W.L.); 4Department of Life Sciences and Chemistry, Jacobs University Bremen GmbH, Campus Ring 1, 28759 Bremen, Germany; e.nevoigt@jacobs-university.de

**Keywords:** pectin-rich residues, sugar beet pulp, non-conventional yeasts, oleaginous yeasts *Rhodotorula toruloides*, *Rhodotorula mucilaginosa*, d-galacturonic acid, l-arabinose, acetic acid

## Abstract

Agro-industrial residues are low-cost carbon sources (C-sources) for microbial growth and production of value-added bioproducts. Among the agro-industrial residues available, those rich in pectin are generated in high amounts worldwide from the sugar industry or the industrial processing of fruits and vegetables. Sugar beet pulp (SBP) hydrolysates contain predominantly the neutral sugars d-glucose, l-arabinose and d-galactose, and the acidic sugar d-galacturonic acid. Acetic acid is also present at significant concentrations since the d-galacturonic acid residues are acetylated. In this study, we have examined and optimized the performance of a *Rhodotorula mucilaginosa* strain, isolated from SBP and identified at the molecular level during this work. This study was extended to another oleaginous red yeast species, *R. toruloides*, envisaging the full utilization of the C-sources from SBP hydrolysate (at pH 5.0). The dual role of acetic acid as a carbon and energy source and as a growth and metabolism inhibitor was examined. Acetic acid prevented the catabolism of d-galacturonic acid and l-arabinose after the complete use of the other C-sources. However, d-glucose and acetic acid were simultaneously and efficiently metabolized, followed by d-galactose. SBP hydrolysate supplementation with amino acids was crucial to allow d-galacturonic acid and l-arabinose catabolism. SBP valorization through the production of lipids and carotenoids by *Rhodotorula* strains, supported by complete catabolism of the major C-sources present, looks promising for industrial implementation.

## 1. Introduction

The implementation of a circular bio-economy based on the efficient bioconversion of agro-industrial residues by selected yeast species/strains is an important societal challenge [1,2,3]. Agro-industrial residues can serve as low-cost raw material for the biotechnology industry, and these organic wastes are sources of carbon, nitrogen and other nutrients for microbial growth and metabolite production while decreasing their negative effects as environmental pollutants [4,5]. Among the agro-industrial residues available, those rich in pectin (e.g., sugar beet pulp (SBP), citrus peels, apple pomace) have potential as feedstocks for the production of biofuels and other industrial bioproducts [3,6]. They are generated in high amounts worldwide from the sugar industry or the industrial processing of fruits and vegetables [7]. Moreover, they are partially pre-treated and have a low lignin content, thus facilitating biomass processing for saccharification and fermentation [8].

Pectin substances are complex heteropolysaccharides and structural components of plant cell walls, having backbone chains of partially methylated α-1,4-linked d-galacturonic acid (homogalacturonan, rhamnogalacturonan II, xylogalacturonan) or d-galacturonic acid and rhamnose (rhamnogalacturonan I), with sugar beet pectin consisting mainly of homogalacturonan, rhamnogalacturonan I and rhamnogalacturonan II [9,10]. The pectin structure complexity depends on variations in methylesterification and acetylation as well as the variety of glycosidic side chains attached to the backbone. These glycosidic side chains are composed of neutral sugars that include l-arabinose, d-xylose, d-galactose, l-rhamnose, l-fucose and d-glucose, among others, linked to d-galacturonic acid and rhamnose units. In particular, arabinose from the branched pectic arabinan side chains of rhamnogalacturonan-I represents a significant percentage of the sugar beet pulp dry matter [9,10]. After the pre-treatment of pectin-rich agro-industrial residues, the enzymatic or acidic hydrolysis of cellulose, hemicellulose and pectin structures allow the release of monomeric sugars that can subsequently be converted by yeasts into a wide range of bioproducts [11,12]. The d-galacturonic acid residues can be methyl-esterified at the C-6 carboxyl group and/or acetylated at C-2 or C-3, and neutralized by ions, like sodium, calcium or ammonium [13]. Therefore, after the hydrolysis of pectin-rich residues, the acetate and methanol generated in the process are present in the hydrolysates and, depending on the concentrations reached, these toxic compounds can inhibit yeast growth and biosynthetic activity [3]. The hydrolysates made from pectin-rich biomass may also contain additional inhibitory compounds such as other weak acids, furan derivatives and phenolic compounds, at different concentrations, depending on the way of processing [14]. Heavy metals and pesticides (fungicides, herbicides and insecticides) used in agriculture are other potential inhibitors of yeast metabolism often present in the hydrolysates [3]. Although the toxicity of some of these compounds at the concentrations present in hydrolysates of pectin-rich residues can be relatively low individually, their toxic effects can be additive or even synergistic, and in combination, they may therefore seriously compromise bioprocesses utilizing this feedstock [14,15,16].

Although *Saccharomyces cerevisiae* is the most important yeast cell factory in the biotechnology industry and the major cell factory platform for the production of bioethanol and other biofuels and chemicals in advanced biorefineries [17,18], the interest in non-*S. cerevisiae* yeast species with higher catabolic and biosynthetic versatility and tolerance to bioprocess-related stresses is gaining momentum [2]. The large and heterogeneous group of nonconventional yeasts includes species/strains with desirable properties for their utilization as cell factories for the synthesis of a wide range of added-value products from a range of carbon sources present in biomass hydrolysates [19,20]. In the case of pectin-rich residues, the efficient and complete utilization by the selected yeast species of the whole mixture of sugar monomers present in the hydrolysates is essential for their economical valorization in industrial biotechnology. Sugar beet pulp hydrolysates contain predominantly the neutral sugars d-glucose, l-arabinose and d-galactose, and the acidic sugar d-galacturonic acid [8,21]. Due to carbon catabolite repression (CCR) [22], these sugars are not simultaneously utilized, which presents another challenge. In fact, CCR leads to inhibition of the uptake and catabolism of the secondary carbon sources, i.e., l-arabinose, d-galacturonic acid, d-xylose, in the presence of the preferred substrate, i.e., d-glucose, prolonging the fermentation time due to the sequential, rather than simultaneous use of all the carbon sources [23,24].

Pectin-rich hydrolysates contain a significant amount of d-galacturonic acid that is neither naturally used by *S. cerevisiae,* nor by other yeast species used in biotechnology, such as *Kluyveromyces marxianus*, *Yarrowia lipolytica* or *Scheffersomyces (Pichia) stipitis*. d-Galacturonic acid is the most challenging sugar to be catabolized, followed by l-arabinose. Several efforts have been dedicated to the development of genetically engineered *S. cerevisiae* strains able to use d-galacturonic acid [25,26,27,28,29] and l-arabinose [30,31] by expression of their catabolic pathways. l-Arabinose is a neutral sugar that is also present in significant amounts in pectin-rich hydrolysates. However, since d-galacturonic acid is more oxidized than the neutral hexose and pentose sugars, the fermentation of d-galacturonic acid to ethanol and CO_2_ is not redox neutral, as opposed to that of d-glucose and related sugars. D-galacturonic acid thus requires more NAD(P)H for its catabolism than generated in its fermentation [32]. Therefore, despite the successful expression in *S. cerevisiae* of enzymes and a membrane transporter comprising a heterologous d-galacturonic acid catabolic pathway using genes from the filamentous fungi *Aspergillus niger* and *Trichoderma reesei,* sufficient regeneration of NADPH has remained a challenge [28,29]. An interesting alternative to *S. cerevisiae* metabolic engineering is the exploitation of nonconventional yeasts for pectin-rich residues’ bioconversion, especially if they are intrinsically able to efficiently use all the carbon sources present in the hydrolysates and to produce relevant added-value compounds [2]. Among those promising yeasts is the basidiomycete red yeast *Rhodotorula toruloides* (formerly known as *Rhodosporidium toruloides*), which has an efficient d-galacturonic acid catabolism [33] and is a potential industrial platform organism for biodiesel and carotenoid production [34]. Industrial production of lipids by oleaginous red yeasts has been considered competitive with plant oil-derived products (e.g., biodiesel), since their production is cleaner and more sustainable [35,36]. *Rhodotorula* strains are also producers of carotenoids used in the food, pharmaceutical and cosmetics industries (e.g., β-carotene, torulene and torularhodin) [37].

In this study, we have examined and optimized the performance of a *Rhodotorula mucilaginosa* strain, isolated from sugar beet pulp and identified at the molecular level during this work, envisaging the full utilization of the carbon sources present in SBP hydrolysate. The study was extended to two promising *Rhodotorula toruloides* strains for bioproduction of lipids and carotenoids [33,34]. SBP hydrolysate supplementation with amino acids was found to be crucial for the efficient catabolism of both d-galacturonic acid and l-arabinose. The dual role of the acetic acid present in SBP hydrolysates as (i) an additional source of carbon and energy for growth and (ii) an inhibitor for growth and metabolism, was also examined. This study provides strong evidence that the concept of SBP valorization through the production of lipids and carotenoids by *Rhodotorula* spp. from the complete catabolism of all major C-sources present in SBP hydrolysates is promising for implementation in an economical biotechnological process.

## 2. Materials and Methods

### 2.1. Isolation and Identification of Rhodotorula mucilaginosa IST 390 from Sugar Beet Pulp

The strain *Rhodotorula mucilaginosa* IST 390 (Instituto Superior Técnico yeast collection) used in this study was isolated from a sample of sugar beet pulp (SBP), a kind gift from the Belgian sugar company Tiense Suikerraffinadarij N.V. Raffinerie Tirlemontoise S.A., obtained in the context of the YEASTPEC (Engineering of the yeast *Saccharomyces cerevisiae* for bioconversion of pectin- containing agro-industrial side-streams) EraNet IB Project. This SBP sample was kept frozen at −20 °C until use. For yeast isolation, 50 g of SBP were mixed with 500 mL of sterile water and chloramphenicol (100 µg/mL) to prevent bacterial growth. This mixture was incubated at 30 °C, with orbital agitation (250 rpm) for one week. After this period, the mixture was serially diluted and plated on YPD agar (Yeast extract 1% (Difco), Peptone 2% (Difco), d-Glucose 2% (Merck), Agar 2% (Merck)) supplemented with chloramphenicol (100 µg/mL). Plates were incubated for 2–4 days at 30 °C. A pink colony identified on the surface of the YPD agar plate was streaked onto a fresh YPD agar plate and incubated under the same conditions to confirm its purity. This yeast isolate, IST 390, was maintained at 4 °C until DNA extraction was performed. For long-term storage, the isolate was preserved at −80 °C in YPD medium containing 15% (*v*/*v*) glycerol.

For the molecular identification of the IST 390 isolate, genomic DNA was extracted using the phenol:chloroform:isoamyl alcohol method [38] and used as a template for the amplification of the D1/D2 domain sequence of the 26S ribosomal DNA (rDNA) and the internal transcribed spacer (ITS) region of rDNA. Polymerase Chain Reaction (PCR) was performed respectively, using primer pairs NL-1 (5′-GCATATCAATAAGCGGAGGAAAAG-3′) and NL-4 (5′-GGTCCGTGTTTCAAGACGG-3′), and ITS1 (5′-TCCGTAGGTGAACCTGCGG-3′) and ITS4 (5′-TCCTCCGCTTATTGATATGC-3′), shown to be effective for the taxonomic identification of yeasts [39]. The PCR protocol consisted of a denaturation step at 98 °C for 30 seconds, followed by 35 cycles of denaturation at 98 °C for 10 s, annealing at 52 °C for 20 s, and extension at 72 °C for 30 s. A final extension was performed at 72 °C for 10 min. The two DNA fragments were purified using NZYGel pure (NZYtech, Lisboa, Portugal) and Sanger-sequenced (Stabvida, Caparica, Portugal) using each corresponding primer. Isolate IST 390 was identified by comparing its D1/D2 and ITS sequences with those deposited in GenBank using the BLAST algorithm from the National Center for Biotechnology Information (NCBI) (http://www.ncbi.nlm.nih.gov/blast, accessed in 7 January 2020). The consensus sequences from D1/D2 and ITS rDNA regions were deposited in GenBank under the accession numbers MW547775 and MW547772, respectively. A phylogenic tree was constructed based on an alignment of available sequences of the D1/D2 domains of the 26S rDNA region from different *Rhodotorula* species, using the Maximum Likelihood method and Tamura–Nei model. The phylogenic tree representation was conducted in MEGA X [40].

### 2.2. Yeast Strains, Growth Media and Conditions

The yeast strains *Rhodotorula mucilaginosa* IST 390, *Rhodosporidium toruloides* PYCC 5615 (Portuguese Yeast Culture Collection), a conjugated strain derived from IFO 0559 × IFO 0880 [41] (Institute for Fermentation, Osaka, Japan) and obtained from the Portuguese Yeast Culture Collection, and *R. toruloides* IFO 0880, a robust host for lipid overproduction and genome-scale metabolic model [41,42] obtained from Belgian Coordinated Collections of Microorganisms, were used in this study.

A growth medium mimicking sugar beet pulp (SBP) hydrolysate was prepared adding either (i) 6.7 g/L of Yeast Nitrogen Base (YNB; including 5 g/L ammonium sulphate and amino acids 10 mg/L of l-histidine, 20 mg/L dl-methionine and 20 mg/L dl-tryptophan) (Fisher Scientific, Waltham, MA, USA), or (ii) 1.7 g/L Yeast Nitrogen Base (without ammonium sulphate and amino acids) (Fisher Scientific) supplemented with 5 g/L of ammonium sulphate or (iii) 1.7 g/L of Yeast Nitrogen Base (without ammonium sulphate and amino acids) (Fisher Scientific), supplemented with 5 g/L of ammonium sulphate and the amino acids in (i). The three media were used to allow different manipulations of the components. They were supplemented with a mixture of carbon sources composed of d-glucose (10 g/L), l-arabinose (12 g/L), d-galactose (3 g/L), d-galacturonic acid (10 g/L) and acetic acid (35 mM, equivalent to 2.1 g/L when the objective was to mimic SBP hydrolysate). The concentration of each carbon source was defined based on its concentration in SBP hydrolysates H11 and H13 (Table 1) and in other batches prepared as described below. Initial medium pH was adjusted to 5.0 using a solution of 10 M NaOH and sterilized by filtration using a 0.2 µm filter (Whatman^®^ Puradisc, Maidstone, UK).

The SBP hydrolysate-based media were prepared using the SBP hydrolysates H11 or H13 prepared as described below and filter-sterilized using a 0.2 µm flow bottle top filter. These media consisted of either: (i) SBP hydrolysate at pH 5.0 (adjusted using 10 M NaOH) supplemented with 5 g/L of ammonium sulphate, (ii) SBP hydrolysate supplemented with 6.7 g/L of Yeast Nitrogen Base (Fisher Scientific), containing, among other components, 5 g/L of ammonium sulphate and 10 mg/L of l-histidine, 20 mg/L dl-methionine and 20 mg/L dl-tryptophan, or (iii) SBP hydrolysate supplemented with 1.7 g/L of Yeast Nitrogen Base (without ammonium sulphate or amino acids), supplemented with different concentrations of ammonium sulphate (1, 2.5 or 5 g/L), and the amino acids present in (ii). These compounds were added to (iii) from stock solutions at the indicated final concentrations, depending on the objective.

For yeast cultivation in SPB hydrolysate-based media (using batches H11 or H13), yeast strains were pre-cultured for 12 h in YPD liquid medium (Yeast extract 1% (Difco), Peptone 2% (Difco), d-Glucose 2% (Merck)) at pH 5.0) (50 mL of medium in 100 mL shake flasks) at 30 °C with orbital shaking (250 rpm). After pre-cultivation, yeast cells were grown in the same medium and conditions and harvested in the mid-exponential phase of growth by centrifugation (5000× *g*, 10 min), washed twice with sterile water and inoculated at an OD_600 nm_ = 4 ± 0.1 or OD_600 nm_ = 8 ± 0.1, in 25 mL SBP hydrolysate (pH 5.0), with or without supplementation with Yeast Nitrogen Base (Fisher Scientific), and with or without ammonium sulphate or amino acids, depending on the experiment. Cultivations were performed in cotton-plugged 50 mL flasks with 25 mL of medium, at 30 °C with orbital shaking (250 rpm). Yeast growth was followed by periodically measuring the optical density during 120–150 h. Culture samples were collected at different time points to determine the extracellular concentration of carbon sources and metabolites.

### 2.3. Effect of Acetic Acid Concentration on Yeast Growth and Carbon Source Catabolism

The effect of acetic acid on the growth profile and carbon source catabolism by *Rhodotorula mucilaginosa* IST 390 and *Rhodosporidium toruloides* PYCC 5615 was tested using the mixed-sugar medium (in YNB with ammonium and amino acids), mimicking SBP hydrolysate but supplemented with increasing concentrations of acetic acid (0, 20 and 40 mM), adjusted to pH 5.0 with 10 M NaOH. Cell inocula with exponentially growing cells were prepared as described above and cells were inoculated at an initial OD_600 nm_ = 4 ± 0.1. Cultivation and sampling were performed as described above.

### 2.4. Preparation of Sugar Beet Pulp Hydrolysate

For the preparation of the SBP hydrolysate H13, a 10 L fermenter reaction vessel was filled with 2.67 kg (wet weight) of pressed SBP from the 2017 sugar beet campaign received from Raffinerie Tirlemontoise, and water was added to a volume of 9 L. The mixture in the fermenter was autoclaved (20 min, 121 °C) and connected to the controller of a Biostat B fermenter (Sartorius, Göttingen, Germany). For enzymatic hydrolysis, 5 mL of Viscozyme L and 5 mL Celluclast (Sigma-Aldrich/Merck, Darmstadt, Germany) were added and the mixture was continuously stirred (4000 rpm) at 40 °C. Prior to use, the commercial enzyme preparations were deprived of low molecular mass solutes by gel filtration chromatography with a PD-10 (Sephadex™ G-25) desalting column (GE Healthcare, Chicago, IL, USA) using 25 mM sodium phosphate buffer pH 6.0 as the eluent, resulting in a 1.4-fold dilution of the original enzyme cocktails, and were filtered with a FiltropurS PES (polyethersolfone) 0.45 µm syringe filter (Sarstedt, Nümbrecht, Nümbrecht, Germany). After 24 h, another 5 mL of each of the Viscozyme L and Celluclast preparations was added followed by further incubation (40 °C, 4000 rpm, 24 h), after which, most of the particulate material was degraded. After centrifugation (20 min, 5400× *g*), the supernatant was filtered using Whatman Folded Filters 597½ (GE Healthcare, Chicago, IL, USA) followed by filter-sterilization with a 0.2 µm flow bottle top filter (Thermo Fisher Scientific, Waltham, MA, USA). The SBP hydrolysate H11 was generated as described for H13 but on 1/10 the scale in a volume of 1 L.

The composition of the different hydrolysates obtained exhibited some variation. The composition of the two different batches (H11 and H13) used in this study is shown in Table 1. These hydrolysates were prepared at the Technical University of Munich laboratory.

### 2.5. Determination of Concentrations of Sugars and Acetic Acid

Culture samples periodically collected were centrifuged (9700× *g*, 3 min) in a microcentrifuge MiniSpin Plus (Eppendorf, Hamburg, Germany) and 100 μL of the supernatant was pipetted into high-performance liquid chromatography (HPLC) vials and diluted with 900 μL of 50 mM H_2_SO_4_. The concentration of carbon sources present in each sample was determined by HPLC (Hitachi LaChrom Elite, Tokyo, Japan), using a column Aminex HPX-87H (Bio-Rad, Hercules, CA, USA) coupled with a ultraviolet(UV)/visible detector (for d-galacturonic acid and acetic acid detection) and refractive index detector (for d-glucose, l-arabinose and d-galactose detection). Ten microliters of each sample were loaded on the column and eluted with 5 mM H_2_SO_4_ as mobile phase at a flow rate of 0.6 mL/min for 30 min. The column and refractive index detector temperature was set at 65 and 40 °C, respectively. The concentration of each sugar and of acetic acid was calculated using calibration curves prepared for each compound.

### 2.6. Assessment of Lipid Production

The production of lipids was assessed by Nile Red staining as previously described [43] with minor modifications. Yeast cells cultivated in 25 mL of SBP hydrolysate supplemented with 1.7 g/L of YNB (without ammonium sulphate and amino acids), 2.5 g/L of ammonium sulphate, and 10 mg/L of l-histidine, 20 mg/L dl-methionine and 20 mg/L dl-tryptophan in 50 mL shake flasks were collected by centrifugation (9700× *g* for 3 min), washed twice with 10 mM potassium phosphate buffer (PBS) (pH 7.0), and the cells were resuspended in 1 mL of PBS with OD_600nm_ adjusted to 1.0. This cell suspension (with OD_600 nm_ = 1) was mixed with Nile Red (Sigma-Aldrich) solution (2.5 µg/mL) (stock solution prepared in dimethyl sulfoxide (DMSO) and acetone (1:1)) followed by microwave treatment (1150 watts power, 30 s). A total of 100 µL, of each cell suspension normalized to OD_600 nm_ = 1 and containing the Nile Red fluorescence dye, was transferred to a black 96-well optical plate (Thermo Fisher Scientific, NY, USA) and relative fluorescence units (RFU) were measured using a FilterMax F5 Multi-Mode Microplate Reader (Molecular Devices) using the excitation and emission wavelengths of 535 and ~625 nm, respectively. Relative neutral lipid content was represented as RFU. Fluorescence measurements were performed for two biological replicates and three technical replicates.

For microscopic observation of lipid droplets inside the yeast cells, 5 µL of the cell suspension previously stained with Nile Red was examined with a Zeiss Axioplan microscope equipped with adequate epifluorescence interface filters. Fluorescence emission was collected with a coupled device camera (Axiocam 503 color; Zeiss, Jena, Germany), and the images were analyzed with ZEN 2 Microscope Software (Zeiss). The exposure time was kept constant among microscopic analyses and intensity measurements were background-corrected.

### 2.7. Assessment of Carotenoid Production

Carotenoid production was assessed after 144 h of yeast cultivation in SBP hydrolysate H13. Cells were collected by centrifugation (5000× *g*, 15 min) and the pellet was lyophilized during 48 h using a freeze dryer (CoolSafe 55–4—ScanVac). The dried pellets were weighed, as dry cell weight (dcw, grams) for total carotenoid content calculation. For carotenoid pigment extraction, disruption of yeast cells was performed by adding 10 mL of acetone and 2 mL of zirconium beads (Sigma-Aldrich; d = 0.5 mm) to lyophilized cell pellets, followed by vigorous mixing during 20 min (at least 2 cycles of mixing) in a vortex (VWR). The suspension was centrifuged (~8000× g, 15 min) and the liquid fraction transferred to a clean 15 mL falcon tube, for acetone evaporation. The carotenoid pellet was dissolved in 1 mL of acetone and the concentration determined by optical absorbance spectroscopy. The absorbance of the colored suspension was measured at λ = 450 nm and the carotenoid content was determined using the formula: $$\mathrm{Total}\;\mathrm{carotenoid}\;\mathrm{content}\;\left(µg\cdot g^{-1}\right)=\frac{A_{total}\times Volume\;\left(\mathrm{mL}\right)\;\times 10^{4}}{A_{1cm}^{1\%}\times sample\;weigth\left(g\right)}\;$$
where $A_{total}$ = absorbance, Volume = total volume of extract (1 mL) and $A_{1cm}^{1\%}$ = 2500, which is the absorption coefficient recommended for mixtures of carotenoids [44].

### 2.8. Statistical Analysis

All experiments were performed at least twice and results from a representative experiment are shown.

## 3. Results

### 3.1. Isolation of Rhodotorula mucilaginosa IST 390 from Sugar Beet Pulp

Sugar beet pulp (SBP) is likely an interesting material for the isolation of different microorganisms since at room temperature, SBP can be degraded by the exo- and endo-polygalacturonases as well as pectin and pectate lyases produced by filamentous fungi, such as *Aspergillus niger* or *Trichoderma reesei*, that hydrolyze the pectin backbone [45]. Moreover, the sugars present in the side chains of the pectin backbone and in pectin-derived oligosaccharides are also released by the action of different glycoside hydrolases and lyases [10]. In this work, a SBP sample suspended in water with chloramphenicol was used as growth medium for the isolation of yeast strains envisaging the full catabolism of the sugars present in pectin, in particular the more challenging sugars d-galacturonic acid and l-arabinose. Among others, strain IST 390 was isolated, identified and used in this study.

The comparison of its D1/D2 and ITS sequences with the sequences deposited in the NCBI database showed that these sequences share 100% identity with the corresponding sequences from *R. mucilaginosa* strains, indicating that strain IST 390 belongs to the *R. mucilaginosa* species, consistent with the phylogenic analysis performed (Figure 1). Recent literature has described the potential of *Rhodotorula* sp. for the biosynthesis of lipids and carotenoids from biomass feedstocks [46,47]. Moreover, *R. toruloides* was demonstrated to have an efficient metabolism for d-galacturonic acid and the underlying molecular basis was elucidated [33]. Since d-galacturonic acid is the major and most challenging sugar present in pectin substances, the isolated *R. mucilaginosa* strain IST 390 was selected for further studies involving the bioconversion of sugar beet pulp hydrolysates.

### 3.2. Differential Utilization of the Carbon Sources Present in SBP Hydrolysate during R. mucilaginosa IST 390 and R. toruloides PYCC 5615 Cultivation 

The SBP hydrolysate H11 includes several carbon sources at different concentrations (Table 1). Since SBP is a not fully defined rich medium, in the first experiments, *R. toruloides* PYCC 5615 and *R. mucilaginosa* IST 390 were cultivated at 30 °C in shake flasks with orbital agitation in hydrolysate H11 without extra nutrient supplementation, but with the pH adjusted from pH 3.1 to pH 5.0 to reduce the toxic effect of the acetic acid (Table 1). Contrary to our expectations, neither d-galacturonic acid nor l-arabinose was used after more than 100 h of cultivation, even though the other carbon sources were all fully used (results not shown). This result led us to supplement the H11 with ammonium sulphate (5 g/L) to avoid a possible nitrogen limitation. However, as with the unsupplemented hydrolysate, the two strains cultivated in the ammonium sulphate-supplemented hydrolysate H11, at 30 °C and inoculated at an initial OD_600 nm_ of 4, were not able to catabolize most of the d-galacturonic acid and l-arabinose during 120 h of cultivation (Figure 2).

The other major carbon sources, d-glucose, d-galactose and acetic acid were rapidly consumed, with galactose being the last. Remarkably, both strains were able to utilize glucose and acetic acid simultaneously, with both being depleted after 48 h. d-galactose consumption was only completed after the exhaustion of d-glucose, presumably due to repression of the d-galactose catabolic pathway (Leloir pathway) by the presence of glucose [48]. d-Galactose utilization was only complete several hours after full acetic acid consumption. After the depletion of these three carbon sources, a marginal decrease in the concentration of l-arabinose and d-galacturonic acid by *R. toruloides* PYCC 5615 was observed (from 30 to 72 h of cultivation), associated with a negligible biomass increase. Their concentrations remained unchanged after 100 h of cultivation (Figure 2A,B).

### 3.3. Sugar Utilization from a Synthetic SBP Hydrolysate Medium Is Affected by Acetic Acid and Improved by Amino Acid Supplementation and Inoculum Increase

Considering the relatively high levels of acetic acid present in SPB hydrolysates, ranging from 30 to 40 mM (or 1.8 to 2.4 g/L), the possibility that its presence could be interfering with the utilization of d-galacturonic acid and l-arabinose was considered. In fact, this carbon source can also be a strong metabolic inhibitor, depending on its concentration, medium pH and the tolerance of the yeast strain. To test this hypothesis, a synthetic SBP medium was prepared in YNB, with 5 g/L of ammonium and supplemented with representative concentrations of the sugars present in the SPB hydrolysates obtained in different batches. Interestingly, the absence of acetic acid from this medium allowed the utilization of d-galacturonic acid and of a significant percentage of the l-arabinose (Figure 3A). Confirming previous results obtained in SBP hydrolysate, when 40 mM of acetic acid were added to the synthetic medium, the utilization of both sugars was abrogated. This response is consistent with the idea that acetic acid affects not only growth kinetics and d-glucose and d-galactose utilization rates but also hinders the capacity of *R. toruloides* PYCC 5615 to catabolize d-galacturonic acid and l-arabinose (Figure 3B).

Given that non-conventional yeasts may require nutrient adjustments to enable the catabolism and conversion of different carbon and nitrogen sources into the desired products, a number of nutrient supplementations were considered. When the commercial YNB (with ammonium sulphate and the amino acids l-histidine (10 mg/L), dl-methionine (20 mg/L) and dl-tryptophan (20 mg/L)) was used, all the above referred sugars were consumed after 50 h of cultivation (Figure 3C). Moreover, when this medium was supplemented with 40 mM of acetic acid, all the sugars were fully consumed after 120 h of cultivation, even though, at a lower rate, accompanying acetic acid induced growth inhibition and the induction of 50 h latency (Figure 3D). 

To assess the effect of increasing concentrations of acetic acid in growth performance and sugars’ catabolism capacity of *R. toruloides* PYCC 5615, this strain was grown in the same synthetic SBP hydrolysate medium in complete commercial YNB (with ammonium and amino acids) supplemented with increasing concentrations (0, 20 and 40 mM) of acetic acid, at pH 5.0. An initial OD_600 nm_ of 4 was used to additionally assess the role of this parameter (Figure 4A–C).

Results indicate that in YNB medium with amino acids and when a higher inoculum concentration was used (initial OD_600 nm_ of 4.0), the negative impact of acetic acid concentrations below 20 mM at pH 5.0 in *R. toruloides* PYCC 5615 growth performance was negligible (Figure 4A,B). Moreover, the use of acetic acid as a carbon source led to a higher final biomass concentration. The increase of acetic acid concentration to 40 mM at pH 5.0 led to a slight extension of lag phase up to 6 hours, and to a decrease in yeast growth rate and in the d-galacturonic acid and l-arabinose consumption rates, when compared to growth in the absence of acetic acid (Figure 4C). Remarkably, the increase of the concentration of the inoculum had a marked positive impact on the yeast performance compared with the results obtained when, under the same conditions, the initial cell concentration was one half (Figure 3D and Figure 4C). In fact, the duration of the lag phase was significantly reduced (from 48 to 8 h) and the utilization of all C-sources was possible after 75 h compared with 100 h of cultivation.

Collectively, these results obtained with media derived from synthetic SBP hydrolysate indicate that the performance of *R. toruloides* PYCC 5615 in SBP hydrolysate is significantly affected by the acetic acid concentration present, and that amino acid supplementation and the increase of the inoculum concentration are important manipulations to be implemented for the bioprocess improvement.

### 3.4. Performance of R. Toruloides PYCC 5615 in SBP Hydrolysate Supplemented with Commercial YNB with Amino Acids and Effect of Ammonium Sulphate Concentration 

Having demonstrated the effect of the presence of amino acids on the catabolism of all the major carbon sources present in synthetic SBP hydrolysate medium, the performance of *R. toruloides* PYCC 5615 was tested in real SBP hydrolysate H11 at pH 5.0 with addition of commercial YNB with amino acids and use of a high inoculum concentration (corresponding to OD_600 nm_ of 8.0) (Figure 5). Results confirm the capacity of this strain to fully use all the carbon sources present after less than 100 h of cultivation, contrasting with the results obtained when amino acids were not supplemented to the hydrolysate and a lower concentration of inoculum was used (Figure 5 compared with Figure 2B). A similar result was observed for *R. mucilaginosa* IST 390 (Appendix A), confirming the beneficial effect of amino acids for the utilization of all carbon sources present in SBP hydrolysates in *Rhodotorula* strains.

Given that the nitrogen demand is determined by the yeast strain used and concentration of C-sources available, and that the carbon-to-nitrogen (C/N) ratio is crucial for the accumulation of lipids in oleaginous yeasts [49,50] and that it was intended to assess the potential of these oleaginous yeasts for SBP valorization, the effect of the concentration of ammonium sulphate supplementation was examined. Although SBP hydrolysates include natural nitrogen sources, in particular primary amino acids and free ammonia [6], the catabolism by *R. toruloides* PYCC 5615 of all C-sources available was not possible in the absence of amino acid supplementation, as shown before. Since the concentration of ammonium sulphate added in former experiments was the one present in commercial YNB (5 g/L) and such a relatively high concentration can affect lipid production yield, the supplementation of SBP hydrolysate H13 with YNB (without ammonium sulphate or amino acids but supplemented with the amino acids present in the commercial medium) and different concentrations of ammonium sulphate (5, 2.5 and 1 g/L) was tested (Figure 6). The results indicate that different initial concentrations of ammonium sulphate may also affect the percentage of consumption of the C-sources available (Figure 6).

The batch of SBP hydrolysate used here, H13, has a lower concentration of all the sugars and of acetic acid compared with H11 (Table 1). This alters the apparent yeast performance compared with other cultivations performed in H11, in which it was easier to complete the use of all the major carbon sources. Apparently, the reduction of ammonium concentration from 5 to 2.5 g/L did not significantly affect the consumption rate or percentage for all the C-sources (Figure 6A,B). However, the addition of only 1 g/L ammonium sulphate limited the percentage of the d-galacturonic acid and l-arabinose sugars used after the consumption of the more easily used C-sources (d-glucose, d-galactose and acetic acid) (Figure 6C). Both d-galacturonic acid and l-arabinose were not fully used. The final biomass concentration attained in the cultivation with 1 g/L ammonium sulphate was slightly lower, suggesting that the lower performance can be, at least partially, attributed to nitrogen limitation.

### 3.5. Preliminary Assessment of Lipid and Carotenoid Production by Three Oleaginous Red Yeast Strains during SBP Bioconversion

The data obtained during this work allowed the identification of a set of experimental conditions leading to the efficient catabolism of the different C-sources present in SBP hydrolysates by *R. toruloides* PYCC 5615. In the last part of the work, the performance of growth, C-sources catabolism, lipid accumulation and carotenoid biosynthesis was also assessed for two other oleaginous red yeast strains (Figure 7). Given that *R. toruloides* PYCC 5615 was derived from *R. toruloides* strains IFO 0559 and IFO 0880 [41], and *R. toruloides* IFO 0880 is considered a robust host for lipid overproduction and genome-scale metabolic model [41,42], this strain was also included in this study as well as *R. mucilaginosa* IST 390 isolated herein (Figure 7).

The assays were carried out in sugar beet pulp hydrolysate H13 supplemented with 1.7 g/L of YNB (without ammonium sulphate or amino acids) and 2.5 g/L of ammonium sulphate (based on the results presented in Figure 6) and the above-described amino acids used to supplement YNB. Among the three oleaginous yeast strains tested, *R. mucilaginosa* IST 390 showed the highest consumption rate of d-glucose, acetic acid and galactose, but all the strains used these C-sources efficiently. The three *Rhodotorula* sp. strains were able to co-consume d-glucose and acetic acid, which is an important advantage in industrial settings when feedstocks containing both carbon sources are used. However, *R. toruloides* PYCC 5615 was the most efficient strain to fully use d-galacturonic acid, with l-arabinose being consumed faster by all three strains. Lipid accumulation in these oleaginous yeast strains was also preliminarily assessed under conditions leading to the full and more rapid utilization of all the C-sources present in SPB hydrolysates (Figure 7A–C). Results suggest that lipids were accumulated in lipid droplets, especially in the *R. toruloides* strains after 48 h of cultivation, when d-glucose, d-galactose and acetic acid were exhausted and continued during the second phase of growth, during which l-arabinose and d-galacturonic acid were co-consumed (Figure 7 and Figure 8). The fluorescence microscopy observations appear to suggest that *R. toruloides* IFO 0880 was the strain that under identical conditions was able to accumulate lipids in larger lipid droplets, while *R. mucilaginosa* IST 390 was less promising in that respect (Figure 8). The cell morphology, as well as the arrangement and size of the lipid droplets, was found to change during yeast cultivation (Figure 8). After 120 h and up to 144 h of cultivation, an intensification of the orange/red color of the yeast cells, presumably related with the increase in carotenoid biosynthesis and associated with lipid droplet turnover, was observed (Figure 8 and Appendix A). In *S. cerevisiae*, the lipid droplet turnover is an intricate process, depending, among others, on phosphorylation events that stimulate lipolytic activity [51]. Under nutrient starvation, lipolytic breakdown is triggered, leading to lipid oxidation to generate cellular energy and channel different metabolites towards synthesis of carotenoids, that are useful to protect the cells from oxidative damage [52,53].

The content in carotenoids is known to depend on the yeast strain, culture medium composition and external physical factors (e.g., light, temperature, aeration, osmotic stress, etc.) [37,54]. The total concentration of carotenoids produced by the three red yeast strains after 144 h of cultivation was assessed. *Rhodotorula mucilaginosa* IST 390 produced the lowest amount of total carotenoids (81 μg/g dcw) (Table 2), while the highest total carotenoid content was observed for *R. toruloides* PYCC 5615 (255 μg/g dry weight).

Although *R. toruloides* IFO 0880 produced lower levels of total carotenoids (158 μg/g dry weight) than *R. toruloides* PYCC 5615 after 144 h of cultivation, this might be because it took more time to use up the total amount of the C-sources. Visual observation of the cell pellets obtained is consistent with this hypothesis since the color intensification in strain IFO 0880 was observed later (after 120 h). This hypothesis may also partially apply to the poorer performance of the *R. mucilaginosa* strain. It also has to be said that the nature of the mixtures of carotenoids produced by *R. mucilaginosa* and *R. toruloides* strains appear to be different, as suggested by the color of the cell pellets: pink-colored pellets were observed for *R. mucilaginosa* IST 390, whereas *R. toruloides* PYCC 5615 and *R. toruloides* IFO 0880 showed a red–orange color [55] (Appendix A). Nevertheless, a chromatographic analysis is required to determine the proportions of individual carotenoids.

## 4. Discussion

Pectin-rich agricultural residues are potential feedstocks for the microbial production of biofuels, bulk chemicals and other added-value compounds. Among them, sugar beet pulp (SBP) and citrus peel waste (CPW) represent a large fraction of those pectin-rich residues [3,56]. Although these residues are currently used for cattle feeding or landfill soil improvement, they are considered interesting alternative feedstocks for biorefineries because they are partially pre-treated and have a low lignin content which facilitates processing to yield a number of potentially metabolizable carbon sources [3,10]. Depending on the concentrations present in the hydrolysates due to the release of acetyl and methyl groups from pectin [57], acetic acid and methanol are potential inhibitors of yeast growth and metabolism, although methanol is not expected to reach concentrations that are inhibitory [58]. Considering the presence of d-galacturonic acid and l-arabinose at very significant amounts in pectin-rich residues hydrolysates, the complete utilization of the complex mixture of carbon sources requires yeast strains with a broad intrinsic catabolic capacity. In recent years, a number of non-conventional yeast species have gained significant attention as promising cell factories for the production of biofuels and added-value compounds based on agro-industrial feedstocks in environmentally friendly bioprocesses [2]. Among them, *Rhodotorula* species/strains have emerged as promising for the natural catabolism of the highly challenging oxidized sugar d-galacturonic acid [33]. However, their performance when cultivated in media with multiple C-sources as those present in pectin-rich residues hydrolysates has not been studied in depth and is the focus of this study. 

In order to find novel yeast isolates with desirable traits to efficiently use the mixture of carbon-sources present in SBP hydrolysates, we explored yeast diversity associated with SBP samples. Even though the SBP sample received was kept frozen, it was possible to successfully isolate and identify different yeast species (e.g., *Rhodotorula mucilaginosa*, *Kluyveromyces marxianus*, *Clavispora lusitanea*, *Cryptococcus laurentii*) from this sample (results not shown). In this work, only the isolated *R. mucilaginosa* strain was tested. However, interestingly, a *C. laurentii* strain was recently successfully used for lipid production from orange peel waste medium [47,59]. *K. marxianus* is also an interesting species due to its thermotolerance and ability to assimilate a wide range of sugars, in particular l-arabinose (but not d-galacturonic acid), and its capacity to efficiently produce bioethanol [60]. For these reasons, future exploration of these other strains obtained from SBP is considered of interest. The potential of the strain *R. mucilaginosa* IST 390 isolated from SBP and of *R. toruloides* PYCC 5615 for the efficient catabolism of the main C-sources present in SBP hydrolysate (d-glucose, d-galactose, acetic acid, d-galacturonic acid and l-arabinose) was examined. The presence of acetic acid in the hydrolysates (30–40 mM or 1.8–2.1 g/L) adjusted at pH 5.0 did not limit the rapid and full utilization of d-glucose, d-galactose and of acetic acid itself from SPB, but, at pH 3.5, growth was abrogated. Independently of the reported capacity of *R. toruloides* for catabolizing d-galacturonic acid [33], the catabolization of this acidic sugar and of l-arabinose was dramatically affected by the presence of acetic acid. This was demonstrated based on the comparison of the consumption profiles of the major C-sources by both species from a synthetic media mimicking SBP hydrolysates either or not supplemented with acetic acid. At higher or lower concentrations, acetic acid is expected to always be present in SBP hydrolysates, therefore limiting the full use of all the C-sources. In fact, since pectin is acetylated in different positions of the d-galacturonic acid molecule, SBP hydrolysis releases acetic acid that accumulates in the hydrolysate, and this is particularly problematic for sugar beet pulp hydrolysates compared with citrus peel hydrolysates [3].

The rapid co-consumption of the d-glucose and acetic acid present in SBP hydrolysates (only supplemented with ammonium sulphate and set up at pH 5.0), by *R. mucilaginosa* and *R. toruloides,* was another interesting observation. This is a much-appreciated trait in industrial bioprocesses because it can lead to the decrease of the production time and energy costs and enhance bioproduct productivity. d-glucose and acetic acid co-consumption is not observed in *S. cerevisiae* wild-type strains because glucose represses acetate metabolism but was also described in the nonconventional food spoilage yeast *Zygosaccharomyces bailii* [61]. The ability to co-consume d-glucose and acetic acid is an additional important trait of *Rhodotorula* species as promising cell factories for lipid and carotenoid production for the valorization of agro-food and forestry residues in which these two C-sources are present. The consumption of d-galactose was found to occur only after d-glucose consumption, suggesting that its catabolism is subject to catabolite repression in *Rhodotorula* species, as in S. *cerevisiae* [48]. In a recent study reporting the growth and carbon sources’ utilization by the oleaginous species *Lipomyces starkeyi* cultivated in unsupplemented sugar beet pulp hydrolysate, the co-consumption of d-glucose and acetic acid was also observed. However, l-arabinose was not assimilated throughout the cultivation time and d-galacturonic acid was not examined [6].

In this study, it was also found, using a synthetic SPB medium, that even in the absence of acetic acid, the efficient full consumption of d-galacturonic acid and l-arabinose by both *R. mucilaginosa* IST 390 and *R. toruloides* PYCC 5615 was only achieved when amino acids (10 mg/L of l-histidine, 20 mg/L of dl-methionine and 20 mg/L dl-tryptophan) were added to the medium or to the real SBP hydrolysate. Our results indicate that amino acid supplementation also enhances the consumption of d-glucose, d-galactose and acetic acid, but, more importantly, may allow the complete use of d-galacturonic acid and l-arabinose. The catabolic pathway for d-galacturonic acid utilization in *R. toruloides* was recently characterized, including enzymes similar to those described in the ascomycetous filamentous fungi *Aspergillus niger* and *Trichoderma reesei* [33]. The pathway that converts d-galacturonate into glycerol in filamentous fungi comprises four different cytosolic enzymes, and two of them (d-galacturonic acid reductase and l-glyceraldehyde reductase) are NADPH-dependent [45]. Therefore, the efficiency of the d-galacturonic acid catabolic pathway in yeasts and filamentous fungi relies on the regeneration of the redox cofactor NADPH, which can occur through the oxidative phase of the pentose phosphate pathway (PPP) [62]. Moreover, in yeasts and filamentous fungi, the pentoses arabinose and xylose are also catabolized through the oxidoreductase pathway which also requires the redox co-factors NADPH and NADH [63,64]. Redox homeostasis is an essential requirement for metabolism maintenance and energy generation due to the involvement of NAD(H) and NADP(H) redox cofactors in several metabolic networks, and in particular, for both d-galacturonic acid and l-arabinose utilization. Therefore, we hypothesize that the supplemented amino acids might be required to counteract the deleterious effects of acetic acid, including on metabolic pathways involved in NADH and NADPH regeneration, thus guaranteeing enough redox potential for the catabolism of d-galacturonic acid and l-arabinose present in SBP hydrolysates by red oleaginous yeasts. Consistent with our hypothesis, it was reported, based on transcriptomic and metabolomic analyses, that the expression of 24 genes involved in NAD(P)/NAD(P)H homeostasis was greatly reduced in acetic acid stressed cells, with these genes being mainly correlated to NAD+ synthesis and redox transformation from NAD(P) to NAD(P)H [65]. Acetic acid-induced intracellular acidification [61] was found to have a crucial influence on the oxidation-reduction potential of specific reductases and dehydrogenases [66] and the redox homeostasis between NAD(P)H and NAD(P) plays a major role in the modification of the metabolic flux in yeast [67]. Additionally, it was found that acetic acid severely reduced adenosine triphosphate (ATP) levels and the gene expression of some nutrient transporters [65,67,68]. Based on transcriptomic and metabolomic analyses, it was also found that acetic acid-induced stress has varying impacts on amino acid metabolism, carbohydrate metabolism and lipid metabolism [65]. Compared with untreated cells, the concentration of all detected amino acids was found to be dramatically reduced in the yeast cells treated with acetic acid at more than one time point and the uptake and biosynthesis of amino acids from glycolysis, tricarboxylic acid (TCA) cycle and other pathways were suppressed upon acetic acid stress [65]. In summary, considering the deleterious effects of acetic acid in maintaining intracellular pH and redox homeostasis and in affecting nutrient uptake, this stress greatly affects the global metabolism, in particular the biosynthesis of amino acids and related carbohydrate metabolism. It is therefore likely that amino acid supplementation may help to counteract its negative effects. Interestingly, in *S. cerevisiae*, there is strong evidence for an altered activity in the oxidative branch of the PPP after methionine supplementation with detection of elevated levels of PPP metabolites and increased abundance of the NADPH-producing enzyme 6-phosphogluconate dehydrogenase [69]. Evidence was also provided supporting the idea that in stress situations, methionine causes an altered activity of the NADPH-producing oxidative part of the pentose phosphate pathway, suggesting that the effects of methionine on stress response are related with an altered activity of this NADP-reducing pathway [69]. In the particular case of our study, amino acid supplementation is, apparently, counteracting the negative effect of acetic acid stress in d-galacturonic acid and l-arabinose catabolism as well as speeding up the utilization of these sugars, even in the absence of acetic acid.

Oleaginous yeasts trigger lipid accumulation when there is an excess of carbon source and other nutrients, particularly nitrogen, limiting growth, achieving more than 20% of their dry biomass as lipids [70]. In this work, a preliminary evaluation of the potential of three *Rhodotorula* strains to produce lipids and carotenoids from SBP hydrolysates under conditions leading to the full utilization of the major C-sources present was conducted. For the total utilization of the C-sources present, the C/N ratio in the hydrolysates was relatively low (approximately 12), compared with values reported in the literature (above 50) [50,71]. For this reason, it is anticipated that there is still room for further optimization of all the process parameters. It should be referred that acetic acid also alters lipid metabolism: long-chain fatty acids were found to accumulate, but the key genes in fatty acid biosynthesis and most of the differently expressed genes involved in lipid metabolism were downregulated [65]. Therefore, this is another issue deserving further studies. At this phase, it is not easy to compare, in the context of this study, the natural tolerance to acetic acid stress of the *Rhodotorula* species examined here with *S. cerevisiae* because natural *S. cerevisiae* strains are not capable of using d-galacturonic and l-arabinose and of lipid overproduction. Remarkably, the time-course microscopic observation of the lipid droplets in the yeast cells indicates that lipid accumulation occurs during the later stages of cultivation when d-galacturonic acid l-arabinose are still present and acetic acid was fully catabolized. Although no systematic and complete optimization of the process was within the focus of this work, collectively, the results obtained strongly support the idea that SBP and *Rhototorula* strains, in particular from the species *R. toruloides,* are promising feedstocks and cell factory platforms for the production of single-cell oils and pigments for energy and food applications in the context of a circular bioeconomy.

## Figures and Tables

**Figure 1 jof-07-00215-f001:**
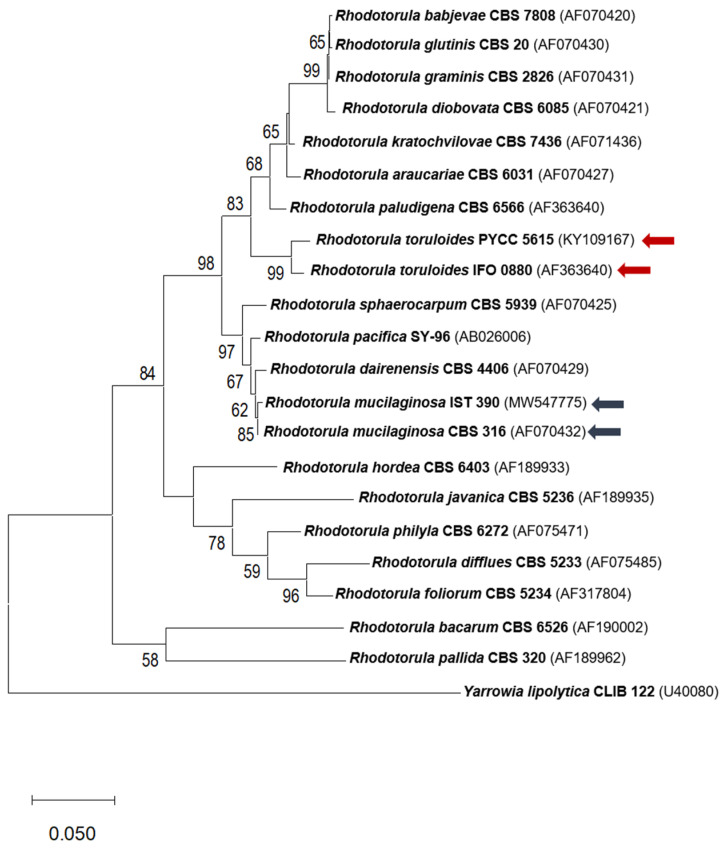
Phylogenetic placement of *Rhodotorula mucilaginosa* IST 390 based on an alignment of sequences of D1/D2 domains of the 26S rDNA region from other *Rhodotorula* species, inferred by using the Maximum Likelihood method and Tamura–Nei model. The scale bar indicates the number of expected substitutions per site. The numbers provided on branches are frequencies with which a given branch appeared in 1000 bootstrap replications. The red and blue arrows indicate *R. toruloides* and *R. mucilaginosa* branches, respectively. The tree was rooted with *Yarrowia lipolytica* CLIB 122 (Collection de Levures d’Intérêt Biotechnologique).

**Figure 2 jof-07-00215-f002:**
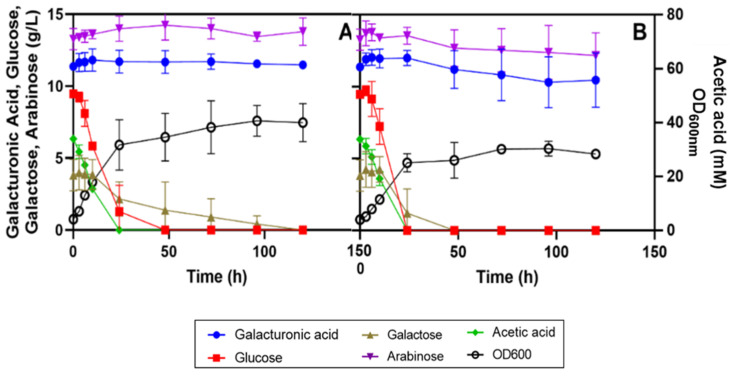
Growth curves and carbon source utilization by (**A**) *Rhodotorula mucilaginosa* IST 390 and (**B**) *Rhodotorula toruloides* PYCC 5615 cultivated in sugar beet pulp (SBP) hydrolysate H11-derived medium. SBP hydrolysate H11 was 5 g/L of ammonium sulphate, adjusted to pH 5.0 and inoculated with an initial optical density at 600 nm (OD_600 nm_) of 4. Yeast cultivations were performed at 30 °C with orbital agitation (250 rpm). The data are means of two independent experiments and bars represent standard deviations (SD).

**Figure 3 jof-07-00215-f003:**
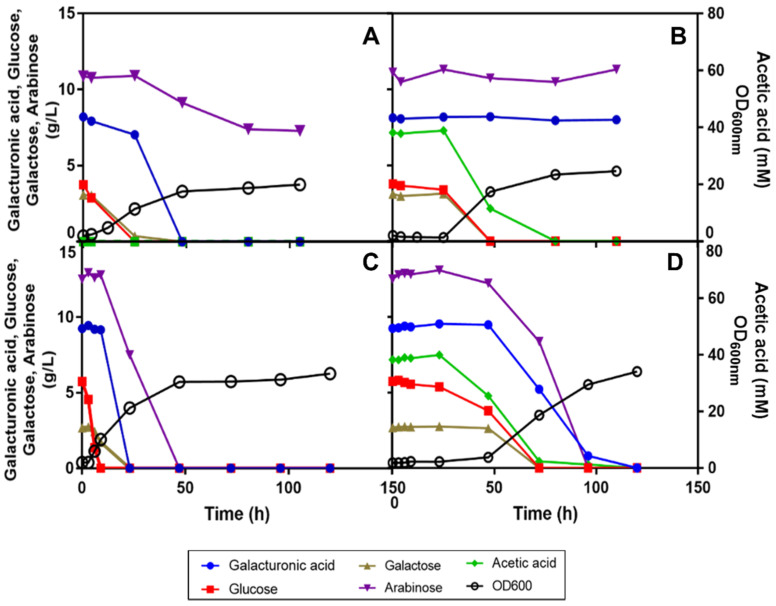
Effect of acetic acid and/or amino acid supplementation of synthetic SPB hydrolysate on the catabolism of different carbon sources by *R. toruloides* PYCC 5615. Growth and carbon sources’ utilization profiles by *R. toruloides* PYCC 5615 when cultivated in: (**A**) mixed-sugar medium (9 g/L d-galacturonic acid, 4 g/L d-glucose, 3 g/L d-galactose, 12 g/L l-arabinose) prepared in Yeast Nitrogen Base (YNB) (with 5 g/L ammonium sulphate), (**B**) the same mixed-sugar medium as in (**A**) supplemented with 40 mM of acetic acid, (**C**) the same mixed-sugar medium as in (**A**) supplemented with amino acids (10 mg/L of l-histidine, 20 mg/L of dl-methionine and 20 mg/L dl-tryptophan) and (**D**) the same mixed-sugar medium supplemented with amino acids as in (**C**) and with 40 mM of acetic acid. Media were adjusted at pH 5.0, cultures were inoculated with an initial OD_600 nm_ of 2 and cultivation was carried out at 30 °C with orbital agitation (250 rpm).

**Figure 4 jof-07-00215-f004:**
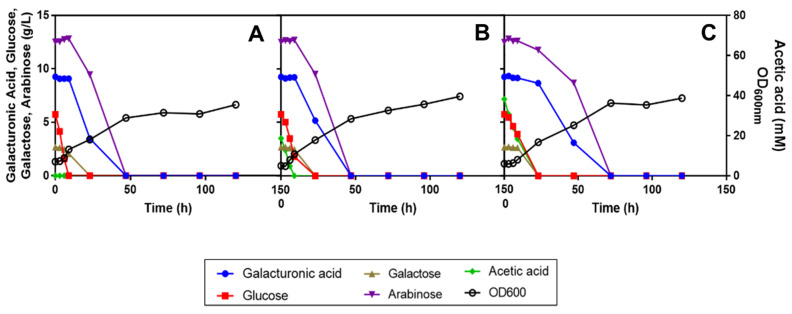
Effect of increasing concentrations of acetic acid (0 (**A**), 20 (**B**) and 40 mM (**C**)) added to the synthetic SBP hydrolysate with the mixture of sugars in commercial YNB (with ammonium sulphate and amino acids) at pH 5.0 in *R. toruloides* PYCC 5615 growth and sugar utilization. All cultures were inoculated with an initial OD_600 nm_ of 4 and cultivation was carried out at 30 °C with orbital agitation (250 rpm).

**Figure 5 jof-07-00215-f005:**
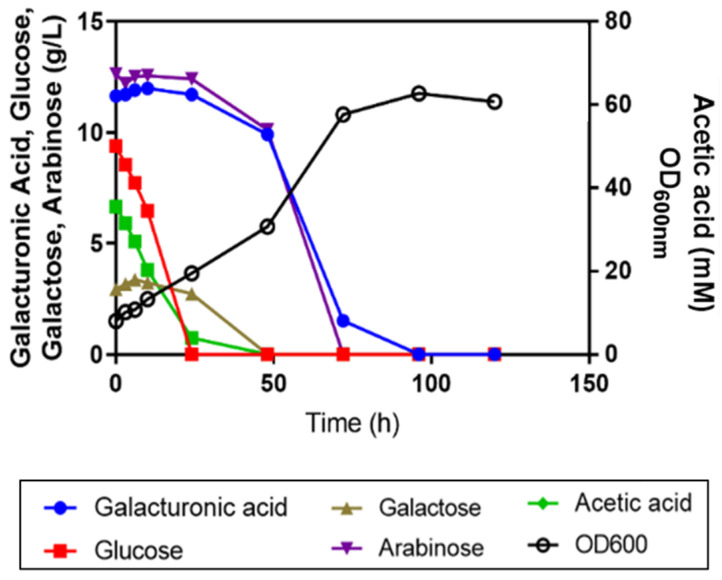
Performance of *R. toruloides* PYCC 5615 in SBP hydrolysate H11 supplemented with commercial YNB with ammonium sulphate and amino acids. Cultivation was performed at 30 °C with orbital agitation (250 rpm). The initial culture OD_600 nm_ was 8.

**Figure 6 jof-07-00215-f006:**
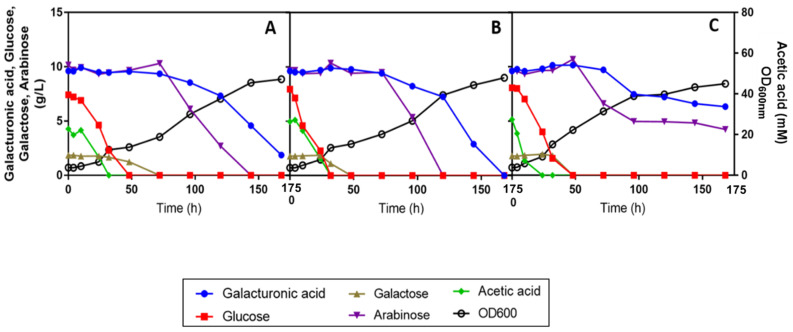
Effect of different initial concentrations of ammonium sulphate, (**A**) 5, (**B**) 2.5 and (**C**) 1 g/L, on the utilization of the different carbon sources present in SBP hydrolysate H13 during the cultivation of *R. toruloides* PYCC 5615. The sugar beet pulp hydrolysate H13 medium at pH 5.0 was prepared in YNB supplemented with the amino acids present in commercial YNB besides the different ammonium sulphate concentrations to be tested. Cultivations were performed at 30 °C, with orbital agitation (250 rpm). Inoculation was with an initial OD_600 nm_ of 4.

**Figure 7 jof-07-00215-f007:**
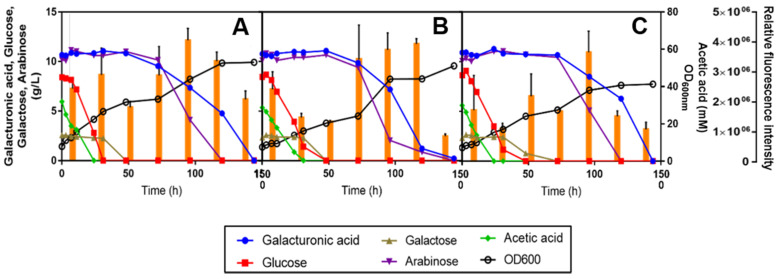
Bioconversion of SBP hydrolysate H13 supplemented with 1.7 g/L of YNB (without ammonium sulphate or amino acids) and with 2.5 g/L ammonium sulphate and the amino acids present in commercial YNB, at pH 5.0, by *R. mucilaginosa* IST 390 (**A**), *R. toruloides* PYCC 5615 (**B**) and *R. toruloides* IFO 0880 (**C**) for lipid production. Cultivations were performed at 30 °C, with orbital incubation (250 rpm). Media were inoculated with cells corresponding to an initial optical density of 8. Lipid production was assessed by Nile Red staining with a normalized cell suspension (OD_600 nm_ = 1) and based on relative fluorescence units (RFU). Orange bars represent average RFU values and standard deviation values are shown.

**Figure 8 jof-07-00215-f008:**
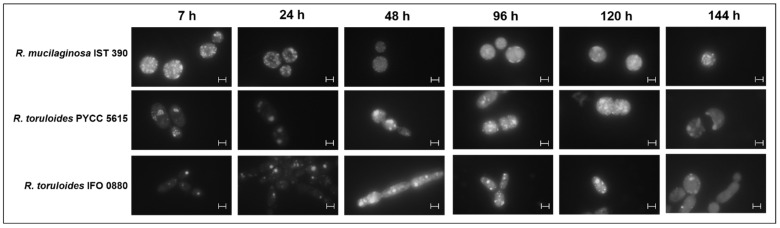
Microscopic observations of *R. mucilaginosa* IST 390, *R. toruloides* PYCC 5615 and *R. toruloides* IFO 0880 cells stained with Nile red fluorescence dye during cultivation in SBP hydrolysate H13-derived medium, as described in the Figure 7 legend. Cells were harvested during the growth curves shown in Figure 7. The arrangement, size and fluorescence intensity of the lipid droplets accumulated inside the cells was examined along yeasts’ cultivation. One of the different pictures taken at the same incubation times is shown as a representative example. The scale bars represent 5 µm.

**Table 1 jof-07-00215-t001:** Composition of the two sugar beet pulp (SBP) hydrolysate batches used in this study.

Scheme	d-Glucose g/L	d-Galactose g/L	l-Arabinose g/L	d-Galacturonic Acid g/L	Acetic Acid mM (g/L)
H11	9.4	4.8	13.1	11.3	33.5 (2.0)
H13	6.7	2.2	9.3	9.4	32.0 (1.9)

**Table 2 jof-07-00215-t002:** Specific production yield of carotenoids (μg/g dry biomass) and volumetric carotenoid concentration (mg/L) produced by the three red yeast strains after 144 h of cultivation in sugar beet pulp hydrolysate H13, as described in the Figure 7 legend.

Yeast Strain	Total Carotenoids	
	(μg/g Dry Biomass)	(mg/L)
*R. mucilaginosa* IST 390	81	1.4
*R. toruloides* PYCC 5615	255	5.4
*R. toruloides* IFO 0880	158	2.8

## Data Availability

Data is available in this article and as Appendix A.

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
