# Peer review of "Complete Utilization of the Major Carbon Sources Present in Sugar Beet Pulp Hydrolysates by the Oleaginous Red Yeasts Rhodotorula toruloides and R. mucilaginosa"

_jof, 2021, doi:10.3390/jof7030215_

Round 1
Reviewer 1 Report
The manuscript by Martins et al examines and optimizes the performance of a Rhodotorula mucilaginosa strain isolated from sugar beet pulp. It describes and discusses its utilization of various carbon sources present in SBP hydrolysate and then analyzes two Rhodotorula toruloides strains for their production of lipids and carotenoids. The study is very well done, methodology appropriate for the subject matter, conclusions supported by the results, all that presented very clearly and a real pleasure to read.
I do not see any serious issues with this work, it is very solid, simple as that. If I had to choose one thing to pick on it would be that the discussion could perhaps use a broader and more precise comparison of carbon usage performance among the strains studied in this manuscript and what has been published in the literature in the past, say, 10 years. This would make it possible to better gauge how much of an improvement/interest they can be for industrial applications. Other than this, I think this work looks great and can be readily published.
Author Response
Reviewer #1 Comments and Suggestions
The manuscript by Martins et al examines and optimizes the performance of a Rhodotorula mucilaginosa strain isolated from sugar beet pulp. It describes and discusses its utilization of various carbon sources present in SBP hydrolysate and then analyzes two Rhodotorula toruloides strains for their production of lipids and carotenoids. The study is very well done, methodology appropriate for the subject matter, conclusions supported by the results, all that presented very clearly and a real pleasure to read.
I do not see any serious issues with this work, it is very solid, simple as that. If I had to choose one thing to pick on it would be that the discussion could perhaps use a broader and more precise comparison of carbon usage performance among the strains studied in this manuscript and what has been published in the literature in the past, say, 10 years. This would make it possible to better gauge how much of an improvement/interest they can be for industrial applications. Other than this, I think this work looks great and can be readily published.
Author’s response:
We appreciate the reviewer comments and agree that the comparison of carbon usage by Rhodotorula spp. is relevant. Unfortunately, for sugar beet pulp hydrolysates, only one recent study is available using the oleaginous yeast Lipomyces starkeyi [6]. However, the sugar utilization performance of this strain was compared with our strains.

Reviewer 2 Report
The presented study by Louis C. Martins and his colleagues is devoted to the important problem of using renewable raw materials. Pectin-rich by-products of sugar beet processing have a number of qualities of an ideal fermentation feedstocks The yeast Saccharomyces cerevisiae, which is most often used in biotechnology for the production of bioethanol, biofuels and other compounds, is not able to convert many components of the polysaccharides of the plant cell wall. The interest in nonconventional yeasts with wide catabolism possibilities makes it possible to expand the range of substrates used. Of particular interest are the red basidiomycete yeast Rhodotorula toruloides, which are able to use D-galacturonic acid and synthesize lipids and terpenes. The authors studied the properties of the yeast strain R. mucilaginosa isolated from sugar beet pulp, adapted to the composition of the nutrient medium, and compared it with the properties of Rhodotorula toruloides from the Yeast Culture Collection.
The revealed dependence of the growth of strains and utilization of carbohydrates in SBP on the concentration of inoculum, ammonium sulfate and amino acid supplementation is important for the industrial use of these strains. Interestingly, the studied strains synthesized various carotenoids. It is obvious that further studies will allow us to clarify the features of carotenoid metabolism in these strains,
Minor note:
Section 3.5 (line 457) should start with a new paragraph.
2 questions:
- What is the reason for choosing histidine, threonine, and methionine as additional amino acids?
- What is the reason for using 1.7 g / L YNB instead of the standard 6.7 g / L?
Author Response
Reviewer #2 Comments and Suggestions
The presented study by Louis C. Martins and his colleagues is devoted to the important problem of using renewable raw materials. Pectin-rich by-products of sugar beet processing have a number of qualities of an ideal fermentation feedstocks The yeast Saccharomyces cerevisiae, which is most often used in biotechnology for the production of bioethanol, biofuels and other compounds, is not able to convert many components of the polysaccharides of the plant cell wall. The interest in nonconventional yeasts with wide catabolism possibilities makes it possible to expand the range of substrates used. Of particular interest are the red basidiomycete yeast Rhodotorula toruloides, which are able to use D-galacturonic acid and synthesize lipids and terpenes. The authors studied the properties of the yeast strain R. mucilaginosa isolated from sugar beet pulp, adapted to the composition of the nutrient medium, and compared it with the properties of Rhodotorula toruloides from the Yeast Culture Collection.
The revealed dependence of the growth of strains and utilization of carbohydrates in SBP on the concentration of inoculum, ammonium sulfate and amino acid supplementation is important for the industrial use of these strains. Interestingly, the studied strains synthesized various carotenoids. It is obvious that further studies will allow us to clarify the features of carotenoid metabolism in these strains,
Minor note:
Section 3.5 (line 457) should start with a new paragraph.
2 questions:
What is the reason for choosing histidine, threonine, and methionine as additional amino acids?
What is the reason for using 1.7 g / L YNB instead of the standard 6.7 g / L?
Author’s response:
A new paragraph was added to “Section 3.5”, as suggested (page 12, lines 465-466).
The amino acids (histidine, methionine and tryptophan) are those included at the referred concentrations in the formulation of commercial Yeast Nitrogen Base (YNB). Their positive impact on D-galacuturonic acid and L-arabinose consumption by Rhodotorula strains was not expected. Their effect was examined and reported.
According to YNB supplier’s recommendations, 1.7 g/L of YNB (without amino acids and ammonium sulphate) should be used, whereas 6.7g/L should be used in the case of YNB (with amino acid and ammonium sulphate). The different concentrations used are related with the different nutrients present in the two mixtures.

Reviewer 3 Report
The manuscript is well written. The results are comprehensive and will be very useful for yeast researches so I recommend the present manuscript to publication in Journal of Fungi after minor revision.
148 – 149 : Please complete the parameters of PCR for ITS and D1/D2 - times and temperatures.
2.7 – Was the 20 minutes of mixing the biomass with the beads in acetone enough to completely extract the carotenoids?
3.1 - Please insert a phylogenetic tree for the new strain of R. mucilaginosa
‘pink-coloured pellets for R. mucilaginosa IST 390 confirming the high percentage of torulene/torularhodin (pink colour) pigment [37], whereas R. toruloides PYCC 5615 and R. toruloides IFO 0880 had a red-orange colour suggesting similar percentages of β-carotene (orange) and torulene/torularhodin carotenoids [54] (Figure S3).’ - I am categorically against such 'conclusions'. The color depends not only on the type of carotenoids, but also on their concentration and proportions of individual carotenoids. Such theories cannot be made without chromatographic analysis. I was working with a strain that was intensively orange and synthesized over 90% beta-carotene.
Author Response
Reviewer #3 Comments and Suggestions
The manuscript is well written. The results are comprehensive and will be very useful for yeast researches so I recommend the present manuscript to publication in Journal of Fungi after minor revision.
148 – 149 : Please complete the parameters of PCR for ITS and D1/D2 - times and temperatures.
2.7 – Was the 20 minutes of mixing the biomass with the beads in acetone enough to completely extract the carotenoids?
3.1 - Please insert a phylogenetic tree for the new strain of R. mucilaginosa
‘pink-coloured pellets for R. mucilaginosa IST 390 confirming the high percentage of torulene/torularhodin (pink colour) pigment [37], whereas R. toruloides PYCC 5615 and R. toruloides IFO 0880 had a red-orange colour suggesting similar percentages of β-carotene (orange) and torulene/torularhodin carotenoids [54] (Figure S3).’ - I am categorically against such 'conclusions'. The color depends not only on the type of carotenoids, but also on their concentration and proportions of individual carotenoids. Such theories cannot be made without chromatographic analysis. I was working with a strain that was intensively orange and synthesized over 90% beta-carotene.
Author’s response:
We appreciate the reviewer’s comments and suggestions. The time and temperatures were added to the parameters of PCR for ITS and D1/D2 amplification, please see page 4 (lines 152-155).
The carotenoids extraction occurred with, at least, two cycles of 20 minutes of mixing to extract the major carotenoids content. The methodology description was corrected (page 6, line 286).
We totally agree that the proportions of each carotenoid and their content must be chromatographically analysed. The sentence was altered to recognize this fact (page 14, line 541-546).
As suggested by the reviewer, a phylogenetic tree was constructed using R. mucilaginosa IST 390 and strains of the Rhodotorula species. The new figure was included in section 3.1 (please see page 7, lines 314-315). The methodology was described in section 2.1 (please see page 4, lines 162-165).
